# Design, Assembly and Control of a Differential/Omnidirectional Mobile Robot through Additive Manufacturing

Erick Axel Padilla-García [1,†], Raúl Dalí Cruz-Morales [2,†], Jaime González-Sierra [3,*,†], David Tinoco-Varela [2,†] and María R. Lorenzo-Gerónimo [1,†]

1 Academy of Robotics Engineering, Polytechnic University of Atlacomulco, Atlacomulco 50465, Mexico; erick.padilla@upatlacomulco.edu.mx (E.A.P.-G.); marialorenzorosaura@gmail.com (M.R.L.-G.)
2 Engineering Department, Superior Studies Faculty-Cuautitlán, National Autonomous University of Mexico, UNAM, Cuautitlán Izcalli 54714, Mexico; rdcruz@comunidad.unam.mx (R.D.C.-M.); dativa19@comunidad.unam.mx (D.T.-V.)
3 Unidad Profesional Interdisciplinaria de Ingeniería Campus Hidalgo, Instituto Politécnico Nacional, Carretera Pachuca—Actopan Kilómetro 1+500, Distrito de Educación, Salud, Ciencia, Tecnología e Innovación, San Agustín Tlaxiaca 42162, Mexico
* Correspondence: jagonzalezsi@ipn.mx
† These authors contributed equally to this work.

**Abstract:** Although additive manufacturing is a relatively new technology, it has been widely accepted by industry and academia due to the wide variety of prototypes that can be built. Furthermore, using mobile robots to carry out different tasks allows greater flexibility than using manipulator robots. In that sense, and based on those above, this article focuses on the design and assembly of a multi-configurable mobile robot that is capable of changing from a differential to an omnidirectional configuration. For this purpose, a sequential mechatronic design/control methodology was implemented to obtain an affordable platform via additive manufacturing which is easily scalable and allows the user to change from one configuration to another. As a proof of concept, this change is made manually. Fabrication, construction, and assembly processes for both structures are presented. Then, a hierarchical control law is designed. In this sense and based on Lyapunov's method, a low-level controller is developed to control the angular speed of the wheels to a desired angular speed, and a medium-level controller controls the robot's attitude to follow a desired Cartesian trajectory. Finally, the control strategies are implemented in both prototype configurations, and through experimental results, the theoretical analysis and the construction of the mobile robot are validated.

**Keywords:** differential-drive mobile robot; omnidirectional mobile robot; additive manufacturing; rapid prototyping

## 1. Introduction

A mobile robot is an automatic machine that can move within a workspace. Mobile robots can be classified into three major areas considering the modes of locomotion and the environment in which they move: aquatic, aerial, and terrestrial [1]. The principal types of terrestrial locomotion are walking, sliding, crawling, climbing, and rotation. Specifically, two terrestrial locomotion rotation modes are rolling and wheels or tracks. The latter represents the most mature mobile robot technology.

The design and build of mobile robots, specifically robots with wheels to perform their motion, have been studied by researchers. These robots are used in industrial applications since they can substitute conveyors, sorting belts, material distribution, and logistics lines. An example is the use of mobile robots for garbage sorting and handling [2]; by using artificial vision and a GPS, the robot can collect or avoid depending on garbage sorting, while an omnidirectional mobile robot is designed to be used as a flexible conveyor; by creating three different layers and different vertical suspension mechanisms, these robots

can be employed to navigate and transport material as a conveyor without the mechanical need to reconfigure it [3]. Another use for designing modular mobile robots is in educational environments, by developing a low-cost robot for workshops with students and using a kit for educational purposes to introduce children to robotics [4]. For these multipurpose objectives, it is essential to design a robot that can be utilized as an omnidirectional mobile robot, which can be conducted in different ways. The easiest one is to use an omnidirectional wheel designed with small rollers that are joined to create a giant wheel that can move in any direction without slipping. Tătar et al. [5] presents multiple structures of these mobile robots. Another low-cost mobile robot can navigate outdoors autonomously using a GPS and a transceiver module to exchange information with other robots and a central station; this robot can perform tasks by itself or in swarm formation [6].

Nowadays, manufacturing techniques are used to fabricate and construct mobile robot variations and can design prototypes more quickly than those used more than 20 years ago. In this sense, the Failure Mode and Effect Analysis (FMEA) technique has been implemented to analyze the risks, reliability, and hazards that can occur while designing and testing the prototype [7]. In the same context, the design and fabrication of a multipurpose surveillance mobile robot is presented using low-cost modules, such as Raspberry Pi and Internet of Things (IoT) devices, to communicate with the robot in different environments [8].

The mobile robot design is not only for typical types of robots but a spherical robot is also fabricated using mechanical design and adaptive estimation of the unbalanced mass that the motion can perform [9]. Although this spherical robot has been studied for many years, its manufacturing is complex. Nevertheless, one can build prototypes of this robot due to the development of different materials manufacturing techniques, including Additive Manufacturing (AM). AM is an emerging methodology focused on increasing productivity and contrary to traditional methods which remove material, this approach creates a product by depositing material layer by layer until all the components are completed [10–12]. This manufactured component's design is developed using the CAD software Shapr3D, version number 5.551.0, and then saved in an STL file where all the information on the material, layers, and properties is stored. This manufacturing method uses PC control, robots, typically a Cartesian robot, a hot table, and a nozzle [12]. The material used for this is plastic, but nowadays, it can be conducted with metal, concrete, and bio-materials for organ printing [13].

AM processes are now used for rapid engineering prototyping because they significantly reduce costs. Therefore, it is used only for prototyping and accurate movements, space, and production tests. By reducing fabrication costs and being socially and environmentally responsible, by using material that can be recycled, this method is also better by reducing material utilization, processing and finishing cost, and fabrication time [14] concerning normal prototype manufacturing. Although rapid prototyping is a very used technique, it has been improved nowadays by using more methods for product development [15,16], and one of these methods is AM, which is a constantly growing technology. Therefore, new trends have emerged, such as hybrid machines, bioprinting, 4D printing [17,18], and the remanufacturing of mechanical products that are out of service [19].

From another point of view, both manipulator robots and mobile robots have been utilized to assist the manufacturing processes. In this sense, in recent years, the term robot-assisted additive manufacturing has emerged, which has brought an improvement in the reduction in production times and flexibility as well as different methodologies and approaches [20,21], such as the iterative, sequential, and concurrent design approach of systems [22], due to the development of materials, processes, and applications in sectors such as aerospace, building construction, automotive and biomedical, among others [23]. For example, Peta et al. [24] conducted experimental studies that determined that the efficiency of robotic production depends on the industrial robot motion parameters to improve manufacturing processes. Xu et al. [25] have proposed a feedforward control method for flexible industrial robots that reduces position and orientation errors for advanced and high-precision manufacturing. Robotic welding systems have played an essential

role in the construction of buildings due to the quality of the welding and the hours of non-stop work. However, programming this type of robot has been quite a challenge; hence, different methods have been developed to tackle these limitations [26]. The survey by Bhatt et al. [27] has demonstrated that manipulator robots with assisted manufacturing can improve part quality and process performance, reduce build time, and fabricate large parts. Outón et al. [28] designed an industrial mobile manipulator that can perform a wide variety of manufacturing processes. Dörfler et al. [29] presents a deep review of research trends and key performance indicators of the use of autonomous mobile robots for building construction, while Lachmayer et al. [30] have developed a contour tracking controller for enabling a mobile robot to perform production tasks while moving on straight wall elements. Similarly, a team of aerial robots for additive manufacturing was introduced for depositing materials during flight and measuring the print quality [31]. Despite the advances that this technology has had, there are still some challenges that have to be identified and analyzed, such as the building of overhang surfaces, part size limitation, limitation in the use of materials, layer misalignment, and poor tolerance accuracy [32].

Based on those mentioned above, and motivated by the vast applications that can be found for these types of mobile robots, the main contributions of this work are:

- To implement a sequential mechatronic design methodology via computing design (CAD, electronics, and control) to obtain a functional multi-configurable mobile robot.
- Employ additive manufacturing techniques for designing and building a differential-drive mobile robot prototype that can transform into an omnidirectional mobile robot by modifying the wheel orientation. The change from one configuration to another is conducted manually as a first step and proof of concept.
- The prototype's design and construction will be validated by implementing two levels of control. In this sense, real-time experiments determine the robot's performance.

To our knowledge, this prototype has yet to be addressed in the literature. It is worth mentioning that a similar approach to the one presented in this article is the one developed by Kladovasilakis et al. [33], where a bio-robotic actuator, which imitates the movement of a human finger, was designed and manufactured with the characteristic that the material could be changed from rigid polymer for industrial applications to thermoplastic elastomer for complete soft robotic applications.

The outline of this work is organized as follows. The methodology is detailed in Section 2, while the design and the instrumentation for both mobile robots are explained in Section 3. Then, Section 4 presents the kinematic model for each structure. The development of the control strategy for both prototypes is described in Section 5, while Section 6 illustrates, through real-time experiments, the performance of both mobile robots. Finally, Section 7 states the conclusions and offers insights into potential future work.

## 2. Methodology

The development of mechatronic systems requires the collaboration of the mechanical, electronic, control, and software domains in a design team. The team can obtain the design and instrumentation by applying a mechatronic design strategy where mechanical, electronic, control, and computer engineering areas must collaborate to obtain a specific mechanism [34]. The typical approach is a sequential design, as shown in Figure 1. The physical system design is obtained in the sequential strategy before designing the control strategy. Later, the combined design is tested to ensure the robot's functionality is simultaneously expected in design and control. This work implements this sequential strategy to design, instrument, assemble, and experimentally evaluate its performance by programming a middle-level controller. Furthermore, a mechanical design on the chassis is proposed in a way that the user can drive an adaptable or configurable robot design, i.e., it is possible to operate the robots in a configuration of a differential-drive robot or a configuration of an omnidirectional robot, with the same proposed mechanical structure.

**Sequential Design Methodology**

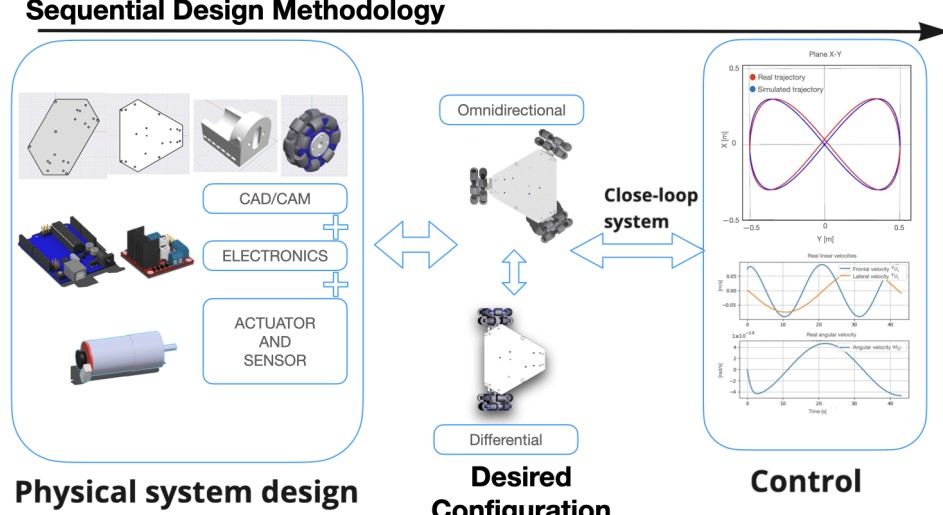

**Figure 1.** Design methodology for the proposed system.

Both differential-drive and omnidirectional mobile robots have a specific kinematics configuration and wheels which is why in this project, the user can modify the chassis to work with one of those desired configurations, for which it will be necessary to simulate the robot at each configuration to verify that the kinematics study is correct. Then, using the same electronic components, an electronic connection scheme can be designed to operate with differential-drive and omnidirectional structures.

After analyzing the robot's simulation, the differential-drive robot's physical assembly must be carried out to transfer what was conducted in the simulations and compare the actual and simulated values. Thus, real-time experiments were conducted with the robot to test the mechanical and electronic configurations. For this purpose, the robot can be monitored to validate the following desired and actual trajectories simultaneously.

A mechanical design is defined as "the process of designing and selecting mechanical components to put them together to achieve a desired function" [35]. Based on this, the design of a configurable differential-drive and omnidirectional robot chassis with the motion characteristics representing each robot is developed based on the sequential integration methodology, described as follows. First, a configurable structural design is proposed via CAD. In a parallel process, the actuators and sensors are selected according to the general dimensions of the proposed configurations, which leads us to present the robot's wheel types. This process can be set via CAD, where the physical design is sought to be obtained in parts according to a prior selection of electronic components. Moreover, this CAD design led us to build the proposed features via CAM (Computer-Assisted Fabrication) by 3D printing the CAD parts. In the same CAD, the assembly and integration of all components are presented, giving the final physical system design. Then, a control strategy is designed to validate the performance and operation.

For many years, the construction of a mobile robot model has had some challenges, such as the high cost and significant electronic devices that do not permit a reduction in the size of each robot. Nevertheless, many different technologies and methods have emerged to create low-cost prototypes that allow us to test them before their final manufacturing, ensuring the proper functioning of the mobile robot. This software has different tools that one can use to create individual parts with material specifications, such as resistance and elasticity, among others. Also, it can create an assembly with each particular component and simulate movement between them. Furthermore, it is possible to simulate forces, torque, and fixed points to obtain mechanical resistance, elongation, deformation, and other mechanical results.

For an optimal mechanical design, first, a prototype was designed on Fusion 360 CAD software from the Autodesk family, where a virtual assembly was conducted to ensure

that all the parts would fit and save material, time, and money when the fabrication was completed.

Mechanical design is a set of a selection of mechanical components and characteristics [35] that are desired for the prototype, which work together to fulfill the machine's goal functionality. The differential-drive mobile robot has a specific configuration; for that reason, one can consider the following assumptions,

- The mechanical structure resistance material can bear three levels with electronic devices in each one.
- The structure dimensions have to follow the electronic devices, wiring, sensors, motors, and moving mechanical elements.
- Electric motors have to be chosen considering all the weight, inertia, and maximum desired speed of the mobile robot.
- The power supply has to be designed to give the necessary current for all the electronic devices, including sensors, electronic boards, motors, and future devices, for at least one complete mobile robotic task.

In the following, a detailed description of each of the elements that make up the mobile robot is given.

### 3. Design and Instrumentation

The design of a configurable differential-drive and omnidirectional robot chassis with the motion characteristics representing each robot is developed based on the sequential integration methodology.

### 3.1. Wheels

Due to this design being used in future mobile robots, omnidirectional wheels have been chosen [36]. The main characteristics of these wheels are summarized as follows:

- Have rollers that allow a sideways motion.
- At least one roller must always be in contact with the ground.
- The geometry of the wheels complies with the equation $\frac{1}{2}x_w^2 + y_w^2 - R^2$, where $R$ is the outer radius and $x_w, y_w$ is the geometric center of the wheel.

In this case, the parameter $R$ is set to 20.5 mm [36]. Seven rollers for each wheel are needed, and the mechanical design is made in CAD. Each wheel has two side caps, one middle cap, rollers, shaft closure, spacer, and shaft coupling. The wheels were built with a 3D Printer ENDER 3 PRO with 180 mm/s printing speed, ±0.1 mm printing precision, and 0.4 mm nozzle diameter [37]. The side caps design has to consider the shafts of the rollers (Figure 2a) and a middle cap inside of the wheel (Figure 2b); each wheel considers two side caps and is made with ABS plastic.

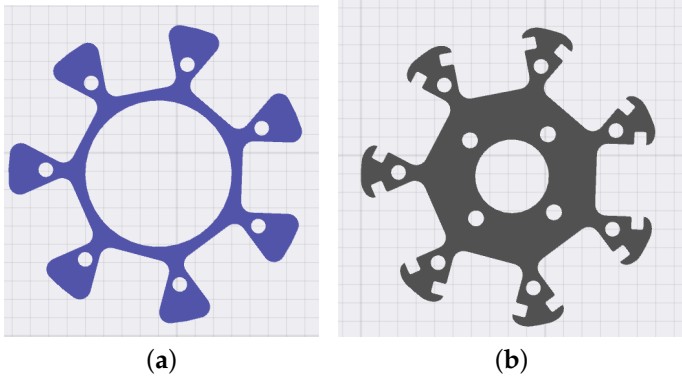

(**a**)        (**b**)

**Figure 2.** Side and middle cap of the wheel. (**a**) The side cap of the wheel. (**b**) The middle cap of the wheel.

The roller (Figure 3a) is the principal part of the omnidirectional wheel and permits the lateral movements of the wheels. Without these rollers, the wheels cannot move sideways. Furthermore, the shaft (Figure 3b) is helpful for the complete assembly of the wheel and putting the motor shaft and joint in the lateral and middle caps.

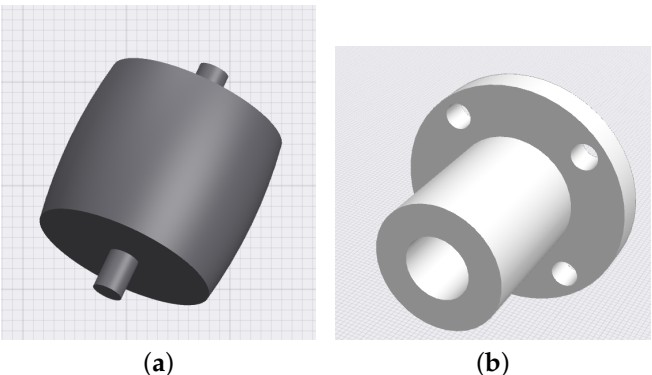

(**a**)                    (**b**)

**Figure 3.** Rollers and shaft closure. (**a**) Rollers. (**b**) Shaft closure.

On the other hand, the spacer (Figure 4a) is utilized to have an identical distance between the lateral caps and the middle cap, while the shaft coupling (Figure 4b) is designed to be coupled to the motor shaft.

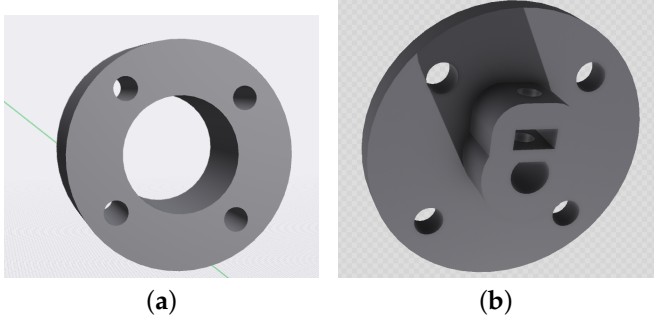

(**a**)                    (**b**)

**Figure 4.** Spacer and shaft closure. (**a**) Spacer. (**b**) Shaft coupling.

Finally, the omnidirectional wheels are assembled using four sides and two middle caps, fourteen rollers, one shaft closure, one spacer, and one shaft coupling, as shown in Figure 5.

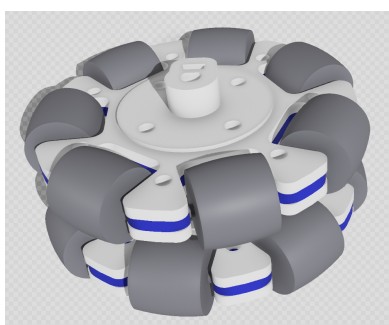

**Figure 5.** Wheel assembly.

### 3.2. Base and Shell Design

The base of the mobile robot is the main piece because all the electronic components, motors, sensors, and parts will be held by it. For that reason, three different levels are proposed. The first level is designed to have the motors and the power battery supply; the

second is designed to hold the electronic devices, sensors, and the idler wheel and the third is only for support and enclosure purposes.

The lower and upper-level bases, illustrated in Figure 6a, are made of ABS plastic printed with a 3D printer. The lower level has 19 holes where the motor supports will fit, and the battery will be held. This same design is used for the upper base as an enclosure but without holes. The middle base level has a triangular shape (Figure 6b), used for the idler wheel to stabilize the robot. Also, this level is made of ABS plastic. The design has 18 screw holes to assemble with the other two levels.

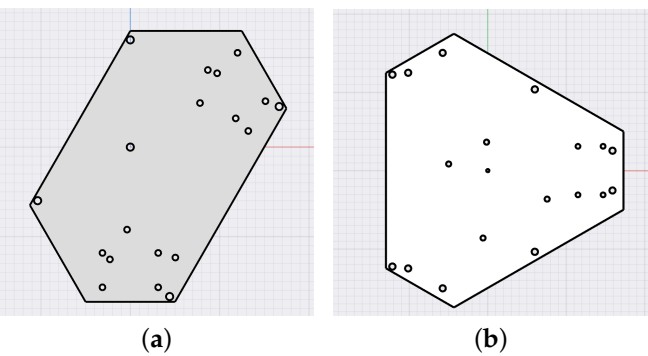

(**a**) (**b**)

**Figure 6.** Lower, upper, and middle-level bases. (**a**) Lower and upper-level bases. (**b**) Middle level base.

The robot's complete base is assembled using two lower-level bases and one middle-level base. All the electronic components for motor control and the idler wheel are set in the middle base, while in the lower-level base, the gear motors bases, gear motor, and wheels are assembled. Finally, the upper level is only used for protection purposes. For the geared motor bases, special supports were designed. Due to a low mass center being needed, this support fits the shape of the gear motor and was made with additive manufacturing with PLA. Figure 7a depicts the design obtained. Furthermore, Figure 7b presents the generic idle wheel that fits in the middle cap with a triangular shape.

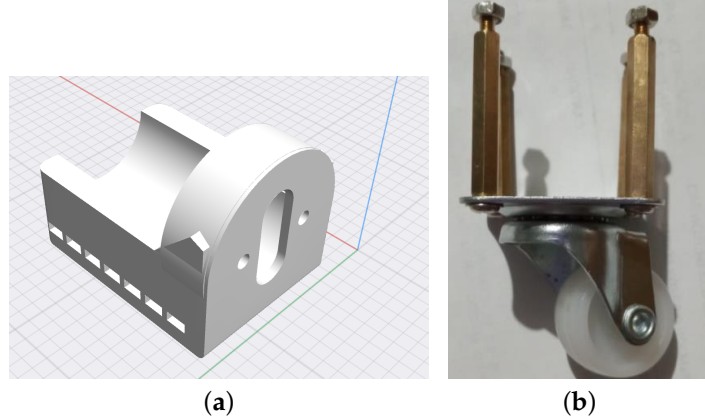

(**a**) (**b**)

**Figure 7.** Geared motor base and idle wheel. (**a**) Geared motor base. (**b**) Idle wheel.

### 3.3. Electronic Design

For the electronic design, four units were proposed to be configured concurrently: the inputs unit, outputs unit, processing unit, and the Source unit, as shown in Figure 8.

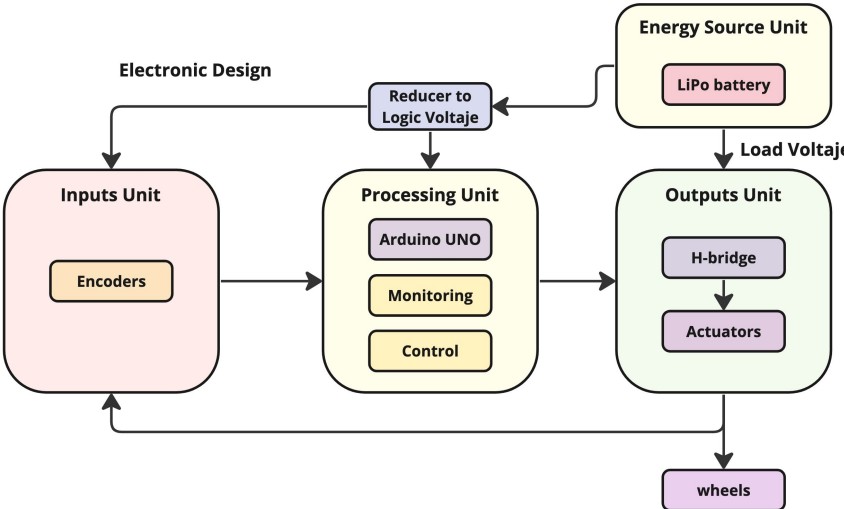

**Figure 8.** Electronic design scheme.

A sensor is required to determine the angular speed of the shaft. Therefore, an incremental rotary-type encoder indicates the motor shaft position, speed, and direction. The incremental encoder uses electrical pulse signals, channel A and channel B, to convert the rotation angle. The pulses of channels A and B depend on each other when the Arduino encoder rotates clockwise or counterclockwise, which is 90 degrees out of phase, as shown in Figure 9.

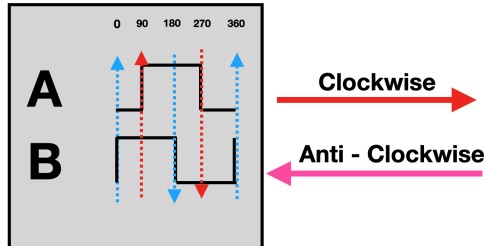

**Figure 9.** Train pulse signals of channel A and channel B during the encoder rotation. Those squared signals can be detected through the rising and falling pulse edges. Red arrows represent the edges of Channel A, while blue arrows represent the Channel B edges.

In an incremental encoder, with both channels A and B, the resolution $R_e$ is the measurement in pulses per revolution (ppr) generated by the encoder for each motor shaft revolution. Assuming that there also exists a gearbox reducer in the motor shaft, the encoder resolution is given by,

$$R_e = t_e \times s_e \times i_R, \tag{1}$$

where $t_e$ is the value of counted ticks per revolution $\times 1$ quadrature, and $i_R$ is the transmission ratio between the motor and the reducer output shaft. For the proposed mobile robot, the encoders are installed in the selected motors of the brand Chihai Motor, model $CHR - GM25 - 370$, where $t_e = 11$ in quadrature $\times 1$, the gearbox reduction ratio of each wheel is 1:45, then $i_R = 45$, and it is possible to read $\times 4$ of a quadrature with this encoder, so $s_e = 4$. Then, the encoder resolution can be computed as $R_e = 1980$ ppr.

### 3.4. Actuator Selection

Depending on the objectives of the robot, different motor types can be used, such as stepper motors, BLDC motors, and DC motors. A low-cost, simple, and quickly widespread controlled motor consists of geared DC motors [38]. The proposed geared DC motors are from Chihai Motor, model $CHR - GM25 - 370$ (Figure 10). The selected motors operate at

12 V, with a rated speed of 350 rpm, a rated current of 1.5 A, a rated load of 3.5 kg·cm, and an average weight of 0.09 kg. The reducer ratio of motors is about 1:45, and at each motor is coupled a rotary incremental encoder, as described above. These motors are powered independently by using an H-bridge motor driver, where these DC motors are commanded from a microcontroller.

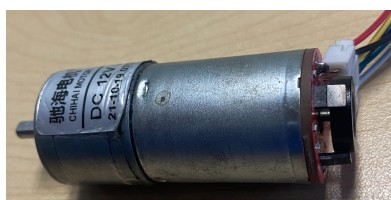

**Figure 10.** DC geared-motors of the robot.

Motor drivers commonly use devices for controlling DC motor types. In this proposal, an L298N module is used to drive the selected DC-geared motors. This module consists of an L298 IC as the motor driver and a five-volt regulator to isolate the logic voltage from the power load voltage. The L298N Module can control up to two DC-geared motors with directional and speed control. In this sense, the differential-drive robot needs a module driver, while for the omnidirectional configuration, two H-bridges are used to handle three motors. These drivers use pulse width modulation (PWM) signals to drive the motors, a commonly used technique to adjust the average voltage to drive DC-geared motors.

### 3.5. Processing and Energy Source Units

The processing unit is responsible for receiving, sending, and monitoring all the information of the mobile robot. It receives the information from input units and processes it to provide the output signals to control the mobile robot.

In this proposal, the Arduino UNO board, which is a low-cost acquisition board to operate the motors, is selected. This board uses an input voltage regulator between 7 and 12 V. Then, it does not require an external regulator to reduce the power voltage of source energy in the processing unit. The power consumption is about 20 mA for 5 volts to control the inputs/outputs signals of the board, for this reason, a LiPo battery of 1000 mAh was selected for logic voltage to energize the board and load voltage to drive the geared motors that can operate the mobile robot for a long time, depending of energy consumption of the control technique and the load of the chassis.

### 3.6. Prototype Assembly

The complete assembly of the mechanical and electronic parts of the prototype is shown in Figure 11 and summarized as follows:

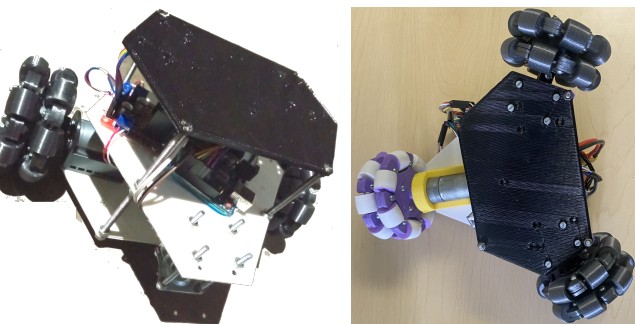

**Figure 11.** Complete prototype assembly.

- The first level has the battery, motors, and supports.
- The second level has electronic devices such as Arduino, L298N H bridge, instrumentation, and cables.

- The third level is designated for all the sensors that will be placed in the future.
- All the levels and motors are assembled by using four 130 mm $\frac{3}{16}''$ screws, twenty 12.5 mm $\frac{3}{32}''$ screws, and four 63 mm $\frac{5}{32}''$ screws for the structure base robot. Also, four 40 mm $\frac{1}{8}''$ screws, twenty 15 mm $\frac{1}{8}''$ screws, and fifty-one nails are used for the shaft of each roller.

## 4. Kinematic Model

This section describes the differential and omnidirectional prototypes. Then, the kinematic model for mobile robots and the relation between the robot velocities and the wheel velocities is presented.

### 4.1. Differential Configuration

In this configuration, the chassis can be moved using the commonly low-cost yellow wheels with a radius of $r_D = 0.0340$ m, as shown in Figure 12. The horizontality of the robot is maintained through a commercial idler wheel type, which is set in the chassis to the necessary measurements for the robot, adapting the required height of installed wheels. Considering that the chassis was designed to be configured as an omnidirectional robot, the omnidirectional wheels, with a diameter of $r_O = 0.0465$ m, can also be used in the differential configuration, where two omnidirectional wheels can be installed in the motors' shafts, as illustrated in Figure 13. The chassis was designed to rotate the motors until they obtained the required configuration. In this case, the axes of the motor shafts should be collinear between them, and the idler wheel has to be adjusted to the required height to ensure the horizontality of the mobile robot. In both assemblies, the instrumentation uses the same components to operate the robot, i.e., two DC-geared motors and one H-bridge motor drive as outputs, two encoders as inputs, one Arduino ONE board as the processing unit, and one external energy source. The only difference between them is the wheel type.

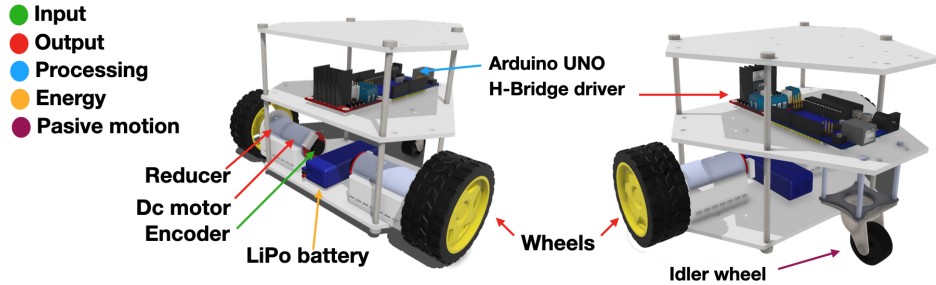

**Figure 12.** Differential robot configuration with yellow wheels.

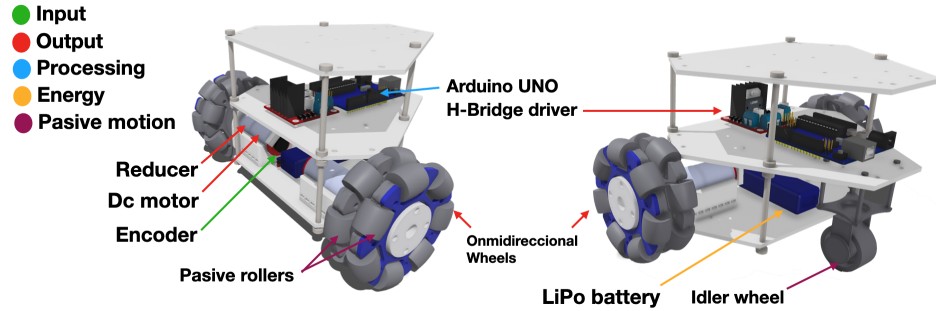

**Figure 13.** Differential robot configuration with omnidirectional wheels.

According to Figure 14, $v_x$ is the wheel longitudinal velocity, $v_y$ the wheel lateral velocity, $\dot{\phi}$, the wheel angular velocity and $r_\ell$ the wheel radius for $\ell = D, O$, with $r_D$ for the yellow standard wheel, and $r_O$ for the omnidirectional wheel. Before proceeding, let us state the following assumptions to determine the robot's velocity.

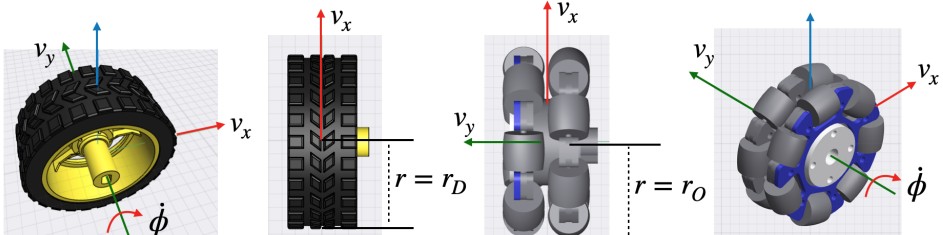

**Figure 14.** Motion of wheels for the differential configuration considering yellow standard or omnidirectional wheels.

**Assumption 1.** *The mobile robot is moving in a horizontal plane. On the other hand, the wheels are rigid enough to maintain a constant radius $r_\ell$, and they are connected rigidly to the chassis. Furthermore, the steering axes of the wheels are orthogonal to the surface.*

**Assumption 2.** *Effects of slipping, sliding, and skidding are neglected during the motion.*

**Remark 1.** *Assumption 2 can take much work to fulfill, especially when the two omnidirectional wheels are used. However, it is assumed that the motor shafts are collinear to ensure pure rolling motion; the wheel longitudinal velocity complies with the following equation*

$$v_x = r_\ell \dot{\phi}, \tag{2}$$

*while the wheel lateral velocity complies with*

$$v_y = 0. \tag{3}$$

Figure 15 depicts the robot's attitude concerning an inertial frame given by the plane $x_I - y_I$. Let $\begin{bmatrix} x_D & y_D \end{bmatrix}^\top$ be the position in the plane of the middle-point of the wheels' axle of the mobile robot concerning the inertial frame; $\theta_D$ is its orientation measured from the horizontal $x_I$ axis. Note, that the robot has its reference frame given by the plane $x_R - y_R$.

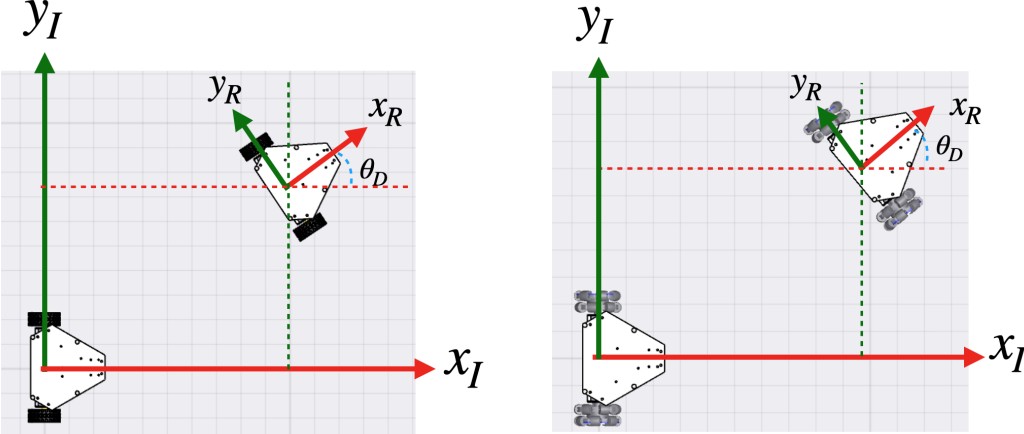

**Figure 15.** Differential robot frame $x_R - y_R$ with respect to the global frame $x_I - y_I$ considering standard wheels and omnidirectional wheels.

The kinematic model of the differential configuration is given by

$$\begin{bmatrix} \dot{x}_D \\ \dot{y}_D \\ \dot{\theta}_D \end{bmatrix} = \begin{bmatrix} \cos\theta_D & 0 \\ \sin\theta_D & 0 \\ 0 & 1 \end{bmatrix} v_D, \tag{4}$$

where $\nu_D = \begin{bmatrix} v_D & \omega_D \end{bmatrix}^\top$ is the vector of the control inputs with $v_D$ as the longitudinal velocity and $\omega_D$ as the angular velocity of the robot. Furthermore, the relation between the wheels' velocities and the body's velocities [39] is defined as

$$\nu_D = D_D(r_\ell, L_D)u_D, \tag{5}$$

where $D_D = r_\ell \begin{bmatrix} \frac{1}{2} & \frac{1}{2} \\ -\frac{1}{L_D} & \frac{1}{L_D} \end{bmatrix}$, $u_D = \begin{bmatrix} \dot{\phi}_L & \dot{\phi}_R \end{bmatrix}^\top$ as the control input vector refers to the angular velocities of the left and right wheel, respectively, and $L_D$ is the distance between the wheels. It is well-known that if one tries to control the coordinates $\begin{bmatrix} x_D & y_D \end{bmatrix}^\top$, the system (4) cannot converge and stabilize by any continuous and time-invariant control law [40]. To overcome this problem, a new control point $h_D$ (Figure 16), located at a distance $a > 0$ from the midpoint of the wheels' axle of the mobile robot, is proposed as

$$h_D = \begin{bmatrix} p \\ q \end{bmatrix} = \begin{bmatrix} x_D + a\cos\theta_D \\ y_D + a\sin\theta_D \end{bmatrix}. \tag{6}$$

The dynamics of (6) is given by

$$\dot{h}_D = J_D(\theta_D)\nu_D, \tag{7}$$

where $J_D(\theta_D) = \begin{bmatrix} \cos\theta_D & -a\sin\theta_D \\ \sin\theta_D & a\cos\theta_D \end{bmatrix}$ is the Jacobian matrix associated to the differential-drive robot. Note, that matrix $J_D$ is non-singular since $\det(J_D) = a \neq 0$; therefore, it is possible to design a control strategy for the point $h_D$.

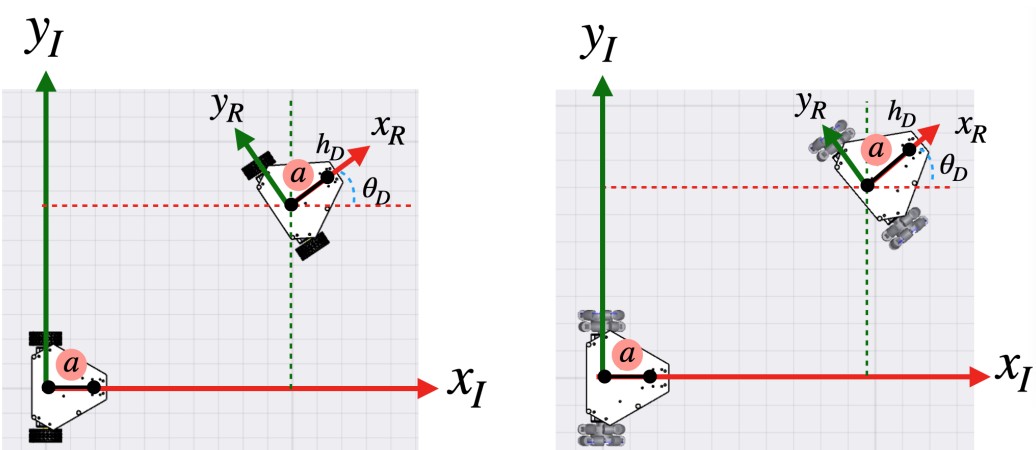

**Figure 16.** Kinematic model of the differential robot with a displacement of the control point.

### 4.2. Omnidirectional Configuration

In this configuration, three DC-geared motors are installed so that the steering axles match the geometric center of the chassis, with 120 degrees between each motor axis, as shown in Figure 17. For this purpose, the three motor supports should be oriented at 120 degrees and screwed, where another motor support and a DC-geared motor with another omnidirectional wheel replace the idler wheel. It is worth mentioning that now two linear velocities are presented for each wheel, $v_{x_i}$ as the wheel longitudinal velocity, $v_{y_i}$ as the wheel lateral velocity, and $u_i$ is the wheel angular velocity for $i = 1, 2, 3$.

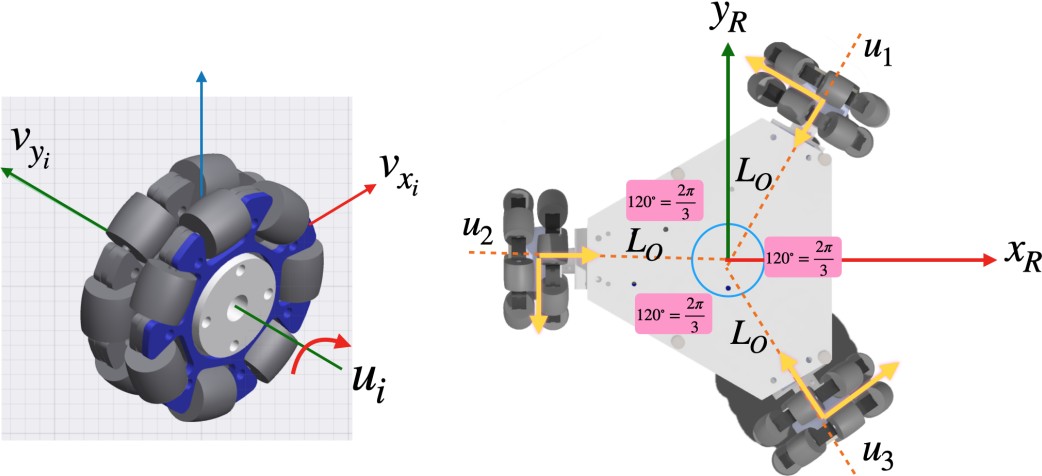

**Figure 17.** Configuration of motors with omnidirectional wheels.

Figure 18 depicts the new assembly configuration, which is described as follows: one encoder is added to the inputs, a motor is added to the outputs, and another H-bridge can be used to drive the third motor.

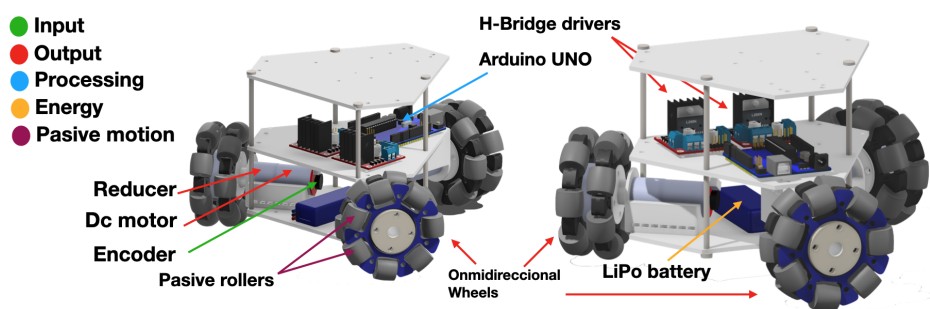

**Figure 18.** Robot configuration with three omnidirectional wheels.

**Assumption 3.** *Note, that this type of robot also considers the Assumptions 1 and 2. Furthermore, it is worth mentioning that the distance $L_O$ is constant and orthogonal to the three wheels, and the three distances match the geometric center of the robot's chassis.*

Figure 19 illustrates the robot's attitude concerning the inertial frame $x_I - y_I$. Let $h_O = \begin{bmatrix} x_O & y_O & \theta_O \end{bmatrix}^\top$ be the attitude of the mobile robot, $\begin{bmatrix} x_O & y_O \end{bmatrix}^\top$ be the position of the geometric center of the robot's chassis and $\theta_O$ the orientation of the robot measured from the horizontal axis $x_I$. The kinematic model is defined as

$$\dot{h}_O = J_O(\theta_O)v_O, \tag{8}$$

where $J_O(\theta_O) = \begin{bmatrix} \cos\theta_O & -\sin\theta_O & 0 \\ \sin\theta_O & \cos\theta_O & 0 \\ 0 & 0 & 1 \end{bmatrix}$ is the Jacobian matrix associated with the omni-

directional robot, and $v_O = \begin{bmatrix} v_{O_x} & v_{O_y} & \omega_O \end{bmatrix}^\top$ is the vector of the control inputs, where $v_{O_x}$ and $v_{O_y}$ are the longitudinal velocity and lateral velocity, respectively, and $\omega_O$ is the angular velocity.

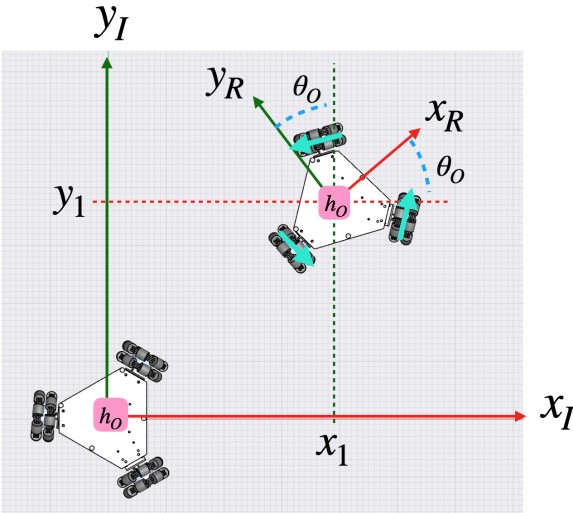

**Figure 19.** Robot frame $x_R - y_R$ with respect inertial frame $x_I - y_I$.

It is considered that each motor speed can be controlled; therefore, the relation between the robot's velocities and the wheels' velocities [39,41] is given by

$$v_O = D_O(r_O, L_O)u_O, \tag{9}$$

where $D_O = r_O \begin{bmatrix} -\frac{1}{\sqrt{3}} & 0 & \frac{1}{\sqrt{3}} \\ \frac{1}{3} & -\frac{2}{3} & \frac{1}{3} \\ \frac{1}{3L_O} & \frac{1}{3L_O} & \frac{1}{3L_O} \end{bmatrix}$, with $u_O = \begin{bmatrix} u_1 & u_2 & u_3 \end{bmatrix}^\top$ as the control input vector.

## 5. Control Strategy

The robot is designed to be controlled in a hierarchical multiprocessor system, as illustrated in Figure 20. In this closed-loop system, the low-level controller is used to control the angular speed of the wheels to a desired angular speed. On the other hand, in the medium-level control, the robot's attitude is controlled to follow a desired Cartesian trajectory. In this sense, for the differential configuration, two controllers are needed in the low-level control scheme to drive real angular velocities $\dot{\phi}_L$ and $\dot{\phi}_R$, with desired speeds, $\dot{\phi}_L^*$ and $\dot{\phi}_R^*$; while for the omnidirectional configuration, three controllers are required for $u_1$, $u_2$, and $u_3$, with desired speeds $u_1^*$, $u_2^*$, and $u_3^*$.

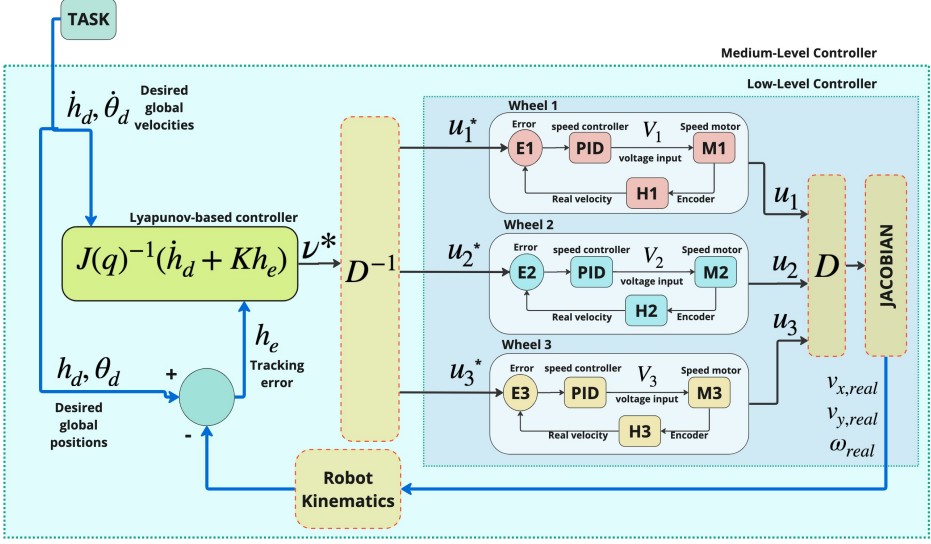

**Figure 20.** Closed-loop control scheme.

*5.1. Low-Level Control*

In the low level, the motion of the wheels is monitored and measured by the encoders and characterized with $H_1$, $H_2$, ..., $H_n$, being the signal conditioners, respectively, to measure the real angular velocities of each wheel. Thus, the Arduino UNO board sets the desired angular velocities to deal with the speed stability. For this purpose, a PID controller has been proposed for each wheel, and it is stated as

$$V_j = K_{p_j} e_j + K_{d_j} \dot{e}_j + K_{i_j} \int_0^{t_f} e_j, \quad j = L, R, 1, 2, 3, \tag{10}$$

where $t_f$ is the time duration of the experiment, $K_{p_j}$, $K_{d_j}$, and $K_{i_j}$ are positive control gains, $V_j$ is the voltage input to control the speed of the motor, and $e_j$ is the tracking error of the motor velocity, defined as

$$e_j = u_j^* - u_j,$$

with $u_j^*$ as the desired angular velocity, and $u_j$ as the measured angular velocity. It is evident that by time $t < t_1$, the error $e_j$ will converge to zero. Therefore, one can assume that $u_j \approx u_j^*$. The problem is the design of the desired angular velocity $u_j^*$.

*5.2. Medium-Level Control*

In the medium-level control, the robot has to track a desired trajectory, which is obtained from the required task, and by using a control technique, the mobile robot is driven to follow it [42]. In this sense, the Lyapunov stability theory is used to obtain the desired inputs of the mobile robot.

Due to the low-level control, one can assume that $u_j \approx u_j^*$; therefore, the desired angular velocities can be obtained from

$$u_\ell^* = D_\ell^{-1} v_\ell^*, \quad \ell = D, O, \tag{11}$$

where $v_\ell^*$ is an auxiliary control, defined as

$$v_\ell^* = J_\ell^{-1}(\dot{\xi}_\ell + K_\ell e_\ell), \quad \ell = D, O, \tag{12}$$

where $\xi_D = \begin{bmatrix} x_d & y_d \end{bmatrix}^\top$ and $K_D = \text{diag}\{k_1, k_2\}$ as the desired trajectory and control gains matrix for the differential robot, respectively; while $\xi_O = \begin{bmatrix} x_d & y_d & \theta_d \end{bmatrix}^\top$ and $K_O = \text{diag}\{k_1, k_2, k_3\}$ are the desired trajectory and control gains matrix for the omnidirectional scheme, respectively, with $k_1$, $k_2$, and $k_3 > 0$. The tracking error $e_\ell$ is defined as

$$e_\ell = \xi_\ell - h_\ell. \tag{13}$$

**Assumption 4.** *Since $u_J \approx u_j^*$, from* (11) *one can conclude that $v_\ell \approx v_\ell^*$.*

*5.3. Stability Analysis*

**Proposition 1.** *Let the control strategy* (12) *in the closed-loop with the system* (7) *for the differential scheme or the system* (8) *for the omnidirectional robot; considering Assumption* 4, *the tracking error* (13), *will converge asymptotically to zero, that is* $\lim\limits_{t \to \infty} e_\ell = 0$, *and the mobile robots will follow the desired trajectory.*

**Proof.** Let us define the following Lyapunov candidate function as

$$\bar{V} = \frac{1}{2} e_\ell^\top e_\ell,$$

whose time-derivative, along the trajectories given in (7) and (8) gives

$$\dot{V} = e_\ell^\top \left[ \dot{\xi}_\ell - J_\ell v_\ell \right]. \tag{14}$$

Considering Assumption 4 and substituting (12) into (14) one obtains

$$\dot{V} = -e_\ell^\top K_\ell e_\ell < 0.$$

Since matrix $K_\ell$ is a definite positive, one can conclude that $\dot{V}$ is a definite negative; the tracking errors converge asymptotically to zero, and the mobile robot follows the desired trajectory. □

## 6. Real-Time Experiments

To test the performance of the proposed design, some experiments on the low-level and medium-level control are employed to show that the configurable design can be used as a differential or omnidirectional robot with the same chassis structure. For this purpose, some trajectory references are given to be followed by using low-level and medium-level control.

### 6.1. Experiment with the Differential Configuration

This experiment focuses on highlighting the importance of medium-level control. Based on those mentioned above, only the low-level controller will be implemented. In this case, the differential robot has to follow a circular trajectory defined as

$$\xi_D = 0.375 \begin{bmatrix} \cos(0.8t) & \sin(0.8t) \end{bmatrix}^\top, \tag{15}$$

$$\dot{\xi}_D = 0.3 \begin{bmatrix} -\sin(0.8t) & \cos(0.8t) \end{bmatrix}^\top, \tag{16}$$

and initial conditions given by $\begin{bmatrix} x_D(0) & y_D(0) & \theta_D(0) \end{bmatrix}^\top = \begin{bmatrix} 0 & 0 & 0 \end{bmatrix}^\top$. From (16), the desired linear velocity can be obtained as $\dot{\xi}_D = \sqrt{\dot{x}_d^2 + \dot{y}_d^2} = 0.3$ m/s, while the desired angular velocity is given by $\omega_D = 0.8$ rad/s. The PID control gains (10) were set to $K_{p_L} = 10$, $K_{p_R} = 8.3$, $K_{d_L} = 0.0001$, $K_{d_R} = 0.0001$, $K_{i_L} = 0.1$, and $K_{i_R} = 0.1$. Figure 21 compares the real and the simulated trajectory performed by the differential robot, with a sample time of $t_s = 0.1$ s. For the simulated trajectory, global linear velocity lies at 0.3 rad/s, which becomes a reference to be reached for the real wheels. The obtained performance of both real linear and angular velocities are given in Table 1.

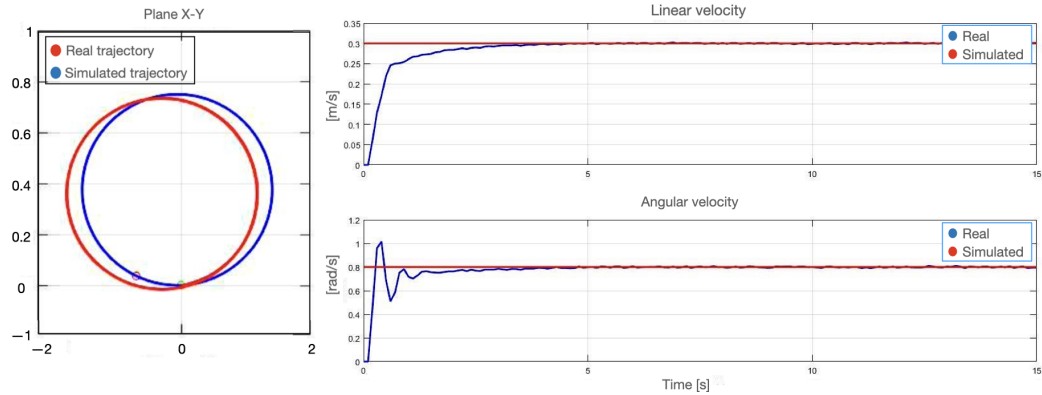

**Figure 21.** Real and simulated trajectory performed by the differential robot.

Note, that the real linear and angular velocities reach the desired linear and angular velocity at $t = 4$ s. Nevertheless, it is worth pointing out that even though the real velocities converge to the desired ones, there is a displacement in the Cartesian trajectory. Based on the above-mentioned, it is clear that the tracking error has to be corrected, and the medium-level control (12) has to be implemented.

**Table 1.** Mobile robot's performance.

| Robot's Velocity | Rise Time [s] | Peak Time [s] | Overshoot [%] | Settling Time 5% [s] |
|:---:|:---:|:---:|:---:|:---:|
| $v$ | 3.7 | - | - | 4.6 |
| $w$ | 0.102 | 0.12 | 25.6 | 2.3 |

*6.2. Experiment with the Omnidirectional Configuration*

In this experiment, both control levels are implemented where the omnidirectional robot has to follow a lemniscate trajectory defined by

$$\xi_O = \begin{bmatrix} 0.3\sin(0.3t) & 0.5\cos(0.15t) & 0 \end{bmatrix}^\top,$$
$$\dot{\xi}_O = \begin{bmatrix} 0.09\cos(0.3t) & -0.075\sin(0.15t) & 0 \end{bmatrix}^\top$$

and initial conditions given by $\begin{bmatrix} x_D(0) & y_D(0) & \theta_D(0) \end{bmatrix}^\top = \begin{bmatrix} 0 & 0 & 0 \end{bmatrix}^\top$. The PID control gains (10) were set to $K_{p_1} = 10$, $K_{p_2} = K_{p_3} = 8.3$, $K_{d_1} = K_{d_2} = K_{d_3} = 0.0001$, and $K_{i_1} = K_{i_2} = K_{i_3} = 0.1$ while the gain controls for (12) were set to $k_1 = k_2 = k_3 = 0.1$. Figure 22 compares the omnidirectional robot's real and simulated trajectory performance and the actual robot's velocities. Note, that the robot can follow the desired Cartesian trajectory when using both level controls.

The obtained tracking errors for the Cartesian trajectory were bounded in the range of $\pm0.025$ m, while the orientation error during the task remains around zero, as shown in Figure 23.

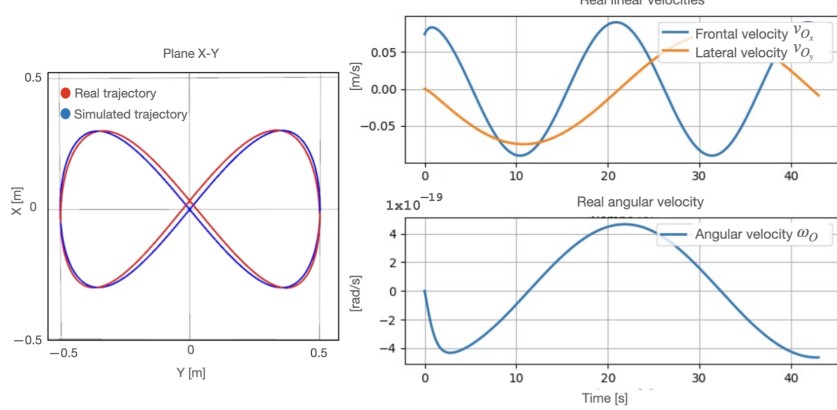

**Figure 22.** Real and simulated trajectory performed by the omnidirectional robot.

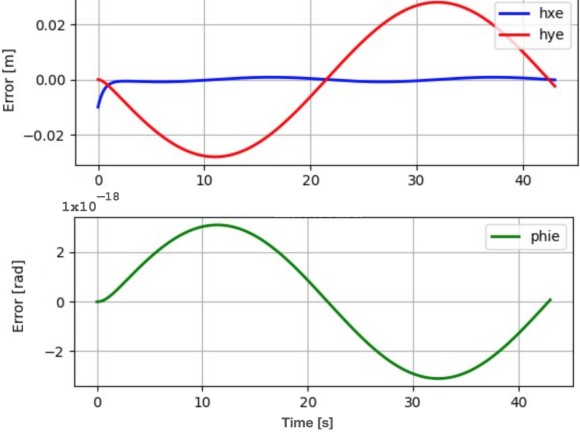

**Figure 23.** Tracking error of Lyapunov controller for the omnidirectional robot.

## 7. Conclusions

AM technology's rise has allowed industry and academia to build prototypes that adapt to their needs. In that sense, the conclusions of this work are summarized as follows:

- Using AM technology and the mechatronic design area allowed us to build a multi-configurable mobile robot that can change from a differential to an omnidirectional configuration by manually orienting the wheels.
- Because the two types of mobile robots addressed in this work have a different kinematic model, a control scheme was proposed to validate their design and construction, consisting of two levels of hierarchy: low and medium levels of control.
- Based on the experiments, it is concluded that, for robots to follow a trajectory in Cartesian space, it is necessary to use two control levels: the low level to control the speed of the wheels and the medium level to control the robot's attitude.

Some open challenges still have to be addressed for future work. For example:

- Implement a mechanism that allows autonomous switching between both configurations.
- Design robust control strategies that deal with external disturbances and communication delays.

**Author Contributions:** Conceptualization, E.A.P.-G. and R.D.C.-M.; methodology, E.A.P.-G. and R.D.C.-M.; software, R.D.C.-M. and M.R.L.-G.; validation, R.D.C.-M., D.T.-V and M.R.L.-G.; formal analysis, R.D.C.-M. and J.G.-S.; investigation, E.A.P.-G. and M.R.L.-G.; resources, E.A.P.-G., R.D.C.-M. and J.G.-S.; data curation, R.D.C.-M. and J.G.-S.; writing—original draft preparation, E.A.P.-G., R.D.C.-M. and M.R.L.-G.; writing—review and editing, J.G.-S. and D.T.-V.; visualization, R.D.C.-M. and J.G.-S.; supervision, R.D.C.-M., J.G.-S. and D.T.-V.; project administration, E.A.P.-G., R.D.C.-M. and D.T.-V.; funding acquisition, R.D.C.-M. and J.G.-S. All authors have read and agreed to the published version of the manuscript.

**Funding:** This work was supported by UNAM PAPIIT IA102323, PAPIME PE104224 and by Instituto Politécnico Nacional through Project SIP: 20240014. The APC was funded by Instituto Politécnico Nacional through Project SIP: 20240014.

**Institutional Review Board Statement:** Not applicable.

**Informed Consent Statement:** Not applicable.

**Data Availability Statement:** Data are contained within the article.

**Conflicts of Interest:** The authors declare no conflicts of interest.

## Abbreviations

The following abbreviations are used in this manuscript:

MDPI    Multidisciplinary Digital Publishing Institute
GPS    Global Positioning System
FMEA    Failure Mode and Effect Analysis
IoT    Internet of Things
AM    Additive Manufacturing
STL    stereolithography
CAD    Computer Assisted Design
CAM    Computer Assisted Fabrication
ABS    Acrylonitrile Butadiene Styrene
PLA    Polylactic Acid
ppr    pulses per revolution
BLDC    Brushless Direct Current
DC    Direct current
PWM    Pulse Width Modulation
PID    Proportional-Integral-Derivative

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
