# Peer review of "Design, Assembly and Control of a Differential/Omnidirectional Mobile Robot through Additive Manufacturing"

_machines, doi:10.3390/machines12030163_

Round 1

Reviewer 1 Report

Comments and Suggestions for Authors

This article is scientifically interesting and overall well written. The topic of the publication fits into current design trends in the additive manufacturing of robots for practical applications. I ask the authors to note the following comments that may improve this paper.

1. Producing robot parts using additive manufacturing is not new. Such solutions are being used more and more often and it is nothing innovative. I ask the authors to write clearly what makes their work unique and what the novelty of the publication is.

2. I ask the authors to refer to literature in a different way. For this purpose, it is worth reading other scientific publications in reputable journals to learn more about this method. You should not write "In [3] an omnidirectional mobile robot..." or "According to [1], mobile robots...".

Rather, the following rule should be used: "Russo et al. [1] presented...." or "Mobile robots can be classified into three major areas considering the modes of locomotion and the environment in which they move: aquatic, aerial, and terrestrial [1]".

3. In the introduction, the authors focused mainly on the description of additive manufacturing of robots, but this description should be supplemented with other important aspects that should be taken into account when designing, manufacturing and operating robots.

Please refer to the literature:

a. https://doi.org/10.1016/j.jmapro.2023.06.063

In this publication, the authors describe the importance of the robot's parameters and movement trajectories for achieving the effectiveness of the robotic process.

b. https://doi.org/10.3390/act12090357 In this publication, the authors propose methods of controlling high-precision robot operation.  

c. https://doi.org/10.1016/j.matpr.2022.03.680 In this publication, the author considers additive manufacturing of robot parts as a step towards sustainable development.   Referring to these publications will allow to look at the research topic in a broader perspective. Even though the publications do not describe exactly the same issues, these topics are universal and worth mentioning.  

4. Line 120 - "we have" - Please do not use this style of sentences, because it is not scientific. Please revise the entire article. Better is: "Nevertheless, nowadays, many different technologies and methods to create.... are used".  

5. Conclusions section is insufficient. In your conclusions, please write what makes your work unique and what your achievements are.  

6. Authors presented comparison between the real and the simulated trajectory (Fig. 21, Fig. 22). Apart from a graphical presentation, are you able to present numerical values of the obtained deviations?

Reviewer 2 Report

Comments and Suggestions for Authors

This is an interesting study that intends to control differential / omnidirectional mobile robots that are 3D printed. There are a  number of issues that the authors need to revise before the manuscript can be considered for publication:

1. In the abstract section, the authors state: using additive manufacturing by utilizing rapid prototyping techniques. Rapid prototyping and additive manufacturing are the same thing. Please revise.

2. The abstract is not complete. There are no results outlined in the abstract and absolutely no details of the work. Please revise.

3. please add a methodology section

4. Please write the conclusions based on qualitative and quantitative values and use bullet points.

Reviewer 3 Report

Comments and Suggestions for Authors

After careful revision of “Design, Assembly and Control of a Differential/Omnidirectional Mobile Robot through Additive Manufacturing”, the reviewer concluded that the paper gives new extended research direction. However, it needs some improvements and major modifications before it goes into publication.

The language is overall good, with few changes required throughout the manuscript.

Please proofread the document carefully for typos, syntax, figures, details, legends, and text format.

Results are missing in abstract. Moreover, Abstract is too small and not structured properly. Abstract should start with importance then problem statement, then researchgap, then methodology then results. And then significance of research.for each of these authors need to add 2/2 lines. 

In Introduction, the line 54-57 needs enriched references such as;

https://doi.org/10.1016/j.aei.2023.102254

doi: 10.1016/j.rcim.2021.102238

doi: 10.1109/TIE.2023.3321997 

In the last paragraph of the introduction section, what is done, how, and what was found should be presented.

The authors should explain how their work differs from the similar in the literature works. This helps highlight the contribution to the field of this work, which is now missing in the work. The novelty of the work should be clearly presented which is very weak in the study.

Material and methods section is fine.

In results and discussion, the authors have not compared their work with previous work. Moreover, authors have used very old references. Authors have provided only 22 references which are not sufficient for a prestigious journal like machines.

Conclusions should contain the result as well. Conclusion section is too short.

Specific comments:

1.     The authors used only 22 references. Please increase upto 30-40 for prestigious journal “Machines”.

2.     Although the topic is origional, relevant to the field but scietific gap has not been addressed properly as no literature has been written. Moreover, the Novelty part is too weak at the end of introduction. Moreover, one figure depicting whole paper schematics and highlighting the novelty will enhance value of paper.

3.     After revising/completing the results/discussion section, make sure conclusions are consistent with the evidence and arguments presented.

4. The reviewer acknowledge that manuscript is scientifically sound, and the experimental design is appropriate to test the hypothesis. However, the structure of paper and lack of results/discussions need to be addressed carefully. Structure for abstract and conclusion, and deep disccusion of results and compare with other studies are missing.

5. Results, significance and limitations are missing in conclusions.

Comments on the Quality of English Language

Minor problems

Round 2

Reviewer 3 Report

Comments and Suggestions for Authors

The authors have really worked hard in revision, paper can be accepted in present form